# Laser Fabrication of Highly Ordered Nanocomposite Subwavelength Gratings

**DOI:** 10.3390/nano12162811

**Published:** 2022-08-16

**Authors:** Yaroslava Andreeva, Alexander Suvorov, Evgeniy Grigoryev, Dmitry Khmelenin, Mikhail Zhukov, Vladimir Makin, Dmitry Sinev

**Affiliations:** 1Institute of Laser Technologies, ITMO University, 197101 Saint Petersburg, Russia; 2Interdisciplinary Resource Center for Nanotechnology of Research Park of SPbSU, Saint-Petersburg State University, 199034 Saint Petersburg, Russia; 3Federal Scientific Research Center “Crystallography and Photonics” RAS, 119333 Moscow, Russia; 4Laboratory of Scanning Probe Microscopy and Spectroscopy, Institute for Analytical Instrumentation RAS, 198095 Saint Petersburg, Russia; 5Institute for Nuclear Energy (Branch), Peter the Great St.Petersburg Polytechnic University, Sosnovy Bor City, 188541 Leningrad Oblast, Russia

**Keywords:** nanocomposite films, nanogratings, TiO_2_, silver nanoparticles, surface plasmon polaritons, direct laser writing

## Abstract

Optical nanogratings are widely used for different optical, photovoltaic, and sensing devices. However, fabrication methods of highly ordered gratings with the period around optical wavelength range are usually rather expensive and time consuming. In this article, we present high speed single-step approach for fabrication of highly ordered nanocomposite gratings with a period of less than 355 nm. For the purpose, we used commercially available nanosecond-pulsed fiber laser system operating at the wavelength of 355 nm. One-dimensional and two-dimensional nanostructures can be formed by direct laser treatment with different scan speed and intensity. These structures exhibit not only dispersing, but also anisotropic properties. The obtained results open perspectives for easier mass production of polarization splitters and filters, planar optics, and also for security labeling.

## 1. Introduction

Periodical structures and nanogratings are advantageous for optics and photonic devices [1,2], sensors [3,4,5], SERS substrates [6,7], security labeling [8], changing of wetting properties [9], antibacterial efficiency [10], and other perspective applications. However, the formation of highly regular gratings with small period is a tricky goal for modern industry due to its physical and optical limitations. Several approaches were proposed, including lithography [11], nanoinscribing [12], surface wrinkling [13], and also laser-based holographic [14,15] and interference [10,16,17,18,19] patterning methods. Although promising, these methods sometimes struggle with providing the necessary flexibility, for instance, in recording the free-shape patterns on curved substrates, the task that can be conducted easily by using the direct laser writing [10,20].

Laser-induced formation of periodical structures on metals probably is one of the most versatile and well studied field of direct laser writing techniques. Various issues of laser-induced periodic surface structures (LIPSS) formation relating to the influence of pulse duration [21], laser processing parameters [21,22,23,24], material [21,22,25], and polarization [23,26,27] were widely discussed. In the paper of [28], it was shown that accumulation effects could drastically affect on the period and regularity of LIPSS. The pulse number reduces LIPPS period due to the change of optical constants and roughness of pretreated area. Another work, [25], emphasizes the key role of the material choice: the more free path length of surface plasmon polariton for the material at the given wavelength, the less regular gratings form. It was also demonstrated that the orientation of polarization vector along scanning direction [29], elliptical polarization [26,27], as well as that a relatively large size of laser spot [25] and ablation [26] can reduce regularity of LIPSS.

The properties of the resulting grating will obviously depend not only on size and shape of structural elements, but also on the material properties. Recently, the perspectives of hybrid materials such as metal-dielectric nanostructures [30,31,32], organic-inorganic materials [33,34], and different nanocomposite materials [8,35,36] have been demonstrated. In some cases, the properties of the periodic lattices (the period of the grating and its height) can be combined with the material properties, thereby expanding the possibilities of their application. It was recently demonstrated that a periodic self-organization of plasmonic nanoparticles within a composite medium can exhibit unique optical effects due to the coupling of lattice modes with plasmon resonances of nanoparticles [8,37]. Similar effects were demonstrated in other studies [32,38,39,40] on periodic arrays of plasmonic and dielectric nanostructures. The optical response of such materials will differ greatly depending on the lattice parameters as well as on the material and dimensions of nanostructures due to the coupling of lattice resonances with electric and magnetic resonances of nanoscale structures. That allows us to use such hybrid structures for coloration, polarization elements, molecular sensing, and nonlinear optics. Thus, there is a great potential hidden behind different hybrid materials.

In this article, we demonstrate that the fast fabrication of highly regular subwavelength gratings consists of titanium dioxide film with small silver nanoparticles. It was shown that nanosecond-pulsed UV exposure within a certain parametric range leads to the formation of one-dimensional and even two-dimensional periodical structures with periods of less than 355 nm. The resulting nanocomposite periodic gratings exhibit anisotropic behavior when viewed in crossed-polarized light: not only the intensity, but also the position of the spectral maxima depends on the rotation angle of the grating. Control throughout the formation of such gratings combined with over processing parameters allows the creation of colorful images with hidden symbols and also paves the way to fast fabrication of various optical elements.

## 2. Materials and Methods

Thin porous film of titanium dioxide impregnated with ions, molecular clusters, and silver nanoparticles (up to 10 nm in diameter) were used as the samples for research. Previously we also investigated silica-based nanocomposit film in terms of nanogratings formation [41]. However, here, we concentrate on titanium dioxide based films since they, being semiconductor, simplify photoreduction processes and find applications in such promising areas as photovoltaics and sensing. The porous TiO_2_ film of about 170–180 nm thick was deposited on a glass substrate by dip coating into the sol–gel solution. Porous films were than soaked in aqueous ammoniacal silver nitrate solution (0.5 M) for 30 min. After that, the films were rinsed by pure water and dried for 24 h in dark. Right before laser treatment, the film was irradiated by UV lamp (λ = 240 nm) to start reduction of silver ions, which ensured the initial absorption of laser irradiation. Similar preparation process of nanocomposite films was described in detail in [42,43].

The gratings were fabricated by commercially available nanosecond-pulsed UV laser system (Laser Center Inc., Saint Petersburg, Russian Federation) The third harmonic of an ytterbium fiber laser (λ = 355 nm) with a pulse duration of 1.5 ns and a pulse repetition rate from 150 to 300 kHz was utilized. The output radiation had a linear polarization and the Gaussian intensity distribution. The beam was focused onto the sample surface with an F-theta lens in a spot with a diameter of d_0_ = 30 μm. Scanning over the surface was carried out using a system of two-axis galvanometric scannators with a variable speed. The study of the surface morphology for the samples before and after treatment was carried out using scanning electron microscopy with the Auriga Laser System-crossbeam SEM-FIB workstation (Carl Zeiss). The surface relief was studied by atomic force microscopy in the semi-contact mode using an NTEGRA Prima (NT-MDT) microscope.

The optical properties of the obtained structures were studied in the regime of crossed polarizers in reflection and transmission modes. For this purpose, a ZEISS Axio Imager A1.m (Carl Zeiss) optical microscope equipped with a rotary object stage was used. To study the spectral properties of the obtained structures, an AvaSpec-ULS4096CL-EVO (Avantes) fiber spectrometer was used. An optical fiber with a diameter of 300 μm was connected to one of the observation channels of the microscope. Thus, the system made it possible to measure the spectra from a region on a sample with a diameter of about 200 μm. To investigate the structure of the nanocomposite gratings, a high-angle annular dark field scanning transmission electron microscopy (HAADF-STEM) and energy-dispersive X-ray analysis were carried out using an electron microscope (Tecnai Osiris, FEI). Cross-sections of the film were prepared using scanning electron microscope (Scios DualBeam, FEI).

## 3. Results

### 3.1. Laser Fabrication of Nanocomposite Gratings

The formation of periodic gratings in a nanocomposite film was carried out by single-step direct laser writing. Focused laser beam scanned the surface line by line with different scan speed and overlap between the neighboring scan lines. The pulse duration in the experiment did not change and was equal to τ = 1.5 ns. To determine the optimal laser processing parameters, under which the formation of periodic nanogratings occur, arrays of squares were recorded with different laser power density and scanning speeds in the range of F = 21–75 mJ/cm^2^ and V*_sc_* = 50–500 mm/s correspondingly, for two values of the pulse repetition rate f = 150 and 300 kHz and overlap n = 100 and 200 lin/mm. To compare experimental results, the parameter number of pulses in the area of influence was used, which was determined as $N=f\cdot n\cdot d_{0}^{2}/V_{sc}$. The resulting arrays were examined by scanning electron microscopy. The development of structures was studied in detail when increasing the laser fluence in the range of F = 21–56 mJ/cm^2^ for constant N = 540 pulses per spot (Figure 1a), and for constant F = 27 mJ/cm^2^ varying N from 54 to 270 pulses (f = 300 kHz, V*_sc_* = 100–500 mm/s) (Figure 1b).

Far from any random combination of laser processing parameters, a periodic relief was formed on the film surface. For instance, for N = 90–540 and laser fluence F = 21–31 mJ/cm^2^ (Figure 1a, Table A1) LIPSS were formed with the average period of 327 ± 10 nm and a grating vector $\vec{g}$ (which is a normal to the grating direction) parallel to the electric field vector $\vec{E}$. For higher repetition rate f = 300 kHz when F = 27–67 mJ/cm^2^ and N = 54–90, the formation of a periodical structures was also observed on the top of the film (Figure 1b, Table A1). It worth knowing that within the investigated range of laser processing parameters, there were no spallation, destruction or ablation of the film observed. The regularity of LIPSS can be expressed via parameter DLOA (the dispersion in the LIPSS orientation angle), which was analyzed by the fast Fourier transform (FFT) of the images. For the obtained structures the average DLOA δθ was 13 ± 4${}^{\circ }$. In some cases, it was possible to reduce DLOA δθ up to 7${}^{\circ }$. The processing of SEM images using FTT also showed the presence of a periodic grating $\vec{G}$ perpendicular to the main relief $\vec{g}$ with the grating vector orientation $\vec{G}⊥\vec{E}$. The period of these high-frequency LIPSS was found to be 186 ± 8 nm. Here and further small letters $\vec{g}$ and d are related to normally oriented LIPSS whereas capital letters $\vec{G}$ and D aree used for abnormally orientated LIPSS. The statistics over the periods of emerging structures are presented in detail in Table A1 in Appendix A.

Studying the surface of the obtained periodic gratings was also carried out by atomic force microscopy (AFM) in the semi-contact mode. Three-dimensional relief maps of the surface of gratings fabricated at N = 540, F = 21 mJ/cm^2^ and 23 mJ/cm^2^ are shown in Figure 2a,b respectively.

The surface profiles of periodic gratings showed that the height of structures with a period of 327 ± 11 nm was about 35 ± 12 nm. At the same time, we did not observe any noticeable changes in the average level of the surface relief. Taking into account the film initial thickness, one can conclude that the formation of periodic structures was not associated with hydrodynamic effects and film melting. It should be noted that periodically distributed inhomogeneities were observed on the top of the ridges, the height of which was about 14 ± 5 nm (profiles 2 and 4 in Figure 2).

Then, transmission electron microscopy was applied to investigate the structure of the nanocomposite film in detail. The thin lamella with the dimentions of about 10 × 2 × 0.1 μm was cut perpendicularly to the grating vector $\vec{g}$ by the focused ion beam. HAADF STEM image and EDX map for titanium (red) and silver (green) are demonstrated in Figure 3. In the vicinity of the grating height minimum, the densified area was obsearved. The film thickness in this area was about 140 ± 10 nm compared to initial value of 175 ± 7 nm.

In contrast to previously reported results [8,44], the formation of periodic nanogratings were not accompanied by the growth of silver nanoparticles and their self-organization.

### 3.2. Optical Properties and Applications

The recorded structures were studied by optical microscopy in a scheme with crossed polarizers. The reflection and transmission spectra of structure (i) (N = 540, F = 21 mJ/cm^2^) and structure (ii) (N = 540, F = 23 mJ/cm^2^) measured in this mode are shown in Figure 4a,b correspondingly.

Both the intensity of the transmitted and reflected light and the position of the spectral maxima depended on the angle of rotation of the structure. The intensity of gratings was maximal when rotated at the angles of 45${}^{\circ }$ and 135${}^{\circ }$. At the codirectional position of the periodic structure vector with one of the axes of the polarizers, the intensity of the spectra was minimal (Figure 4a,b). The reflection and transmission spectra of structure (i) exhibited pronounced peaks, the position of which depended on the rotation angle. With the rotation from 0 to 90 degrees, a reflection maximum was observed at a wavelength of 628 nm, while the spectra also contained a peak at a wavelength of about 515 nm. With further rotation through the angles from 105 to 180 degrees, the peak in the region of 515 nm became the most intense. Similar behavior were also observed in the transmission spectra of structure (i). The visually observed color of the structures during their rotation also changed from blue at 0–90${}^{\circ }$ to green at 105–180${}^{\circ }$.

The spectra of structure (ii) were significantly different. For rotation angles other than 0${}^{\circ }$, 90${}^{\circ }$, and 180${}^{\circ }$, the reflection spectrum had a maximum at wavelengths of 500–550 nm. For 0${}^{\circ }$, 90${}^{\circ }$, and 180${}^{\circ }$ the reflection spectrum was uniform and much less intense. The picture is completely different for the transmission spectra of structure (ii). Regardless of the angle of rotation, the structures have a relatively low transmittance; however, the effect of shifting the spectral minimum to the short-wavelength region with a change in the angle of the structure rotation by 0–90 and 105–180 degrees was also traced.

The described behavior in the reflection and transmission of the gratings indicate that the recorded reliefs have both polarization and dispersion properties. In addition, the degree of manifestation of these properties is seriously affected by the orientation of the structures relative to the polarizer axis.

Examples of possible applications of the described nanocomposite gratings are shown in Figure 5. The combination of different laser exposure modes made it possible to record various color images with hidden security symbols. For instance, one of the sectors on the color wheel was made as the periodic structure (i). When observed in transmitted and reflected light, each sector has its own color. However, if the structure is illuminated from a small angle, this sector exhibits dispersion properties. Such a color hologram can be used as a hidden security label, while observing the surface in crossed polarizers introduces an additional degree of protection against counterfeiting.

## 4. Discussion

The formation of structures with a period of the order of the wavelength, perpendicular to the polarization of radiation, is associated with the excitation of a surface plasmon polaritons (SPP) at the interface between the nanocomposite film and air. It is important to note that previously SPP excitation on TiO_2_:Ag films were reported only for femtosecond laser processing [45,46]. For nanosecond- and longer irradiation, interference of waveguide modes with incident laser beam was the main proposed mechanism of periodic structures formation [8,44].

The conditions for the appearance of such a wave at the boundary between the active media and air are well known and met when Re $\varepsilon_{1}$ < −1, where $\varepsilon_{1}$ is the dielectric permittivity of the medium. Such conditions can be reached than the free carrier density in the media increases due to laser induced excitation, therefore TiO_2_ film acts as metallic.

The free carrier density $N_{e}^{*}$ required for the SPP excitation of pure TiO_2_ can be assumed according the Drude model [47] using following equation (see Appendix B for details):(1)$$N_{e}^{*}>\frac{\left(Re\left(\varepsilon_{film}\right)+1\right)\left(\omega^{2}+\gamma^{2}\right)\varepsilon_{0}\mu_{opt}^{*}m_{e}}{e^{2}}$$
where Re($\varepsilon_{film}$ = 8.8384 [48]) is the real part of dielectric permittivity for TiO_2_ film, ω is the laser frequency, $\gamma ∼10^{14}c^{-1}$ is the damping constant [49], $\varepsilon_{0}$ is the dielectric permittivity, $\mu_{opt}^{*}∼0.01$ is the optical effective mass of free carriers [49], $m_{e}$ is the free electron mass, and *e* is the electron charge.

Thus, the estimated critical free carrier density for SPP excitation in the TiO_2_ film—air interface is found to be $n_{e}^{*}$ > 6.8 × 10${}^{20}$ cm^−1^. For pure TiO_2_ film laser fluence of about 0.3–0.5 J/cm^2^ is needed [49]. However, in the case of processing a composite film, laser radiation is absorbed not only on titamium dioxide but also on silver nanocrystals and nanoparticles. Moreover, the irradiaton wavelength 355 nm (3.49 eV) is close to the bandgap for TiO_2_ film (3.43–3.49 eV [50]), so the absorption efficiency is relatively high. Therefore, electron density increases due to the photoexcitation of TiO_2_ matrix and photoemission of hot electrons from silver nanoparticles into the environment. The latest process is even more effective due to the photoreduction of Ag ions [43] and enhanced absorption of silver for the given wavelength. The contribution of photoexcitation and photoemission processes can be magnified by accumulation effects for a large number of pulses per spot (more than 500) acting at high repetition rate [28]. In addition, due to the impregnation conditions of the film, the concentration of nanoclusters and silver particles on the surface is higher than in the bulk of the film, which also leads to rapid metallization of the interface. These factors create conditions for the excitation of SPP.

The SPP, in turn, interferes with the incident radiation and forms a standing wave and a grating $\vec{g}\left|\right|\vec{E}$, the period of which is defined as d=λ/η, where λ is the wavelength, η is the real part of the complex refractive index of the air-TiO_2_:Ag interface for SPP. Thus, for structures with a period of d = 327 nm, η is 1.086.

Periodic modulation of the field near the boundary affects the conditions for heating, thus modifying the film, which in turn influences the absorption of subsequent pulses and changes surface morphology. The process of periodic grating $\vec{g}$ formation occurs through the positive feedback loop of surface absorption enhancement via growing relief height. Therefore, accumulation effects for large number of pulses lead to the decrease of LIPSS formation threshold and affect their regularity.

Formation of a periodically modulated relief with a period of about 186 ± 8 nm with $\vec{G}⊥\vec{E}$ occurred on the ridges and in the grooves of the major periodical relief. Such structures are associated with the modes called wedge and channel plasmon-polariton modes [51]. These modes are localized near the interface between two media, propagate along extended groves or ridges in two opposite directions, and then mutually interfere. The period of formed structures is defined as D=λ/2ξ, where λ is the wavelength, ξ is the real part of the complex refractive index of the interface air– TiO_2_:Ag for channel (wedge) plasmon-polariton modes. For structures such as ξ = 0.954, which is almost equal to η for SPP.

It have been previously demonstrated in [25] that the less SPP decay length of the material $L_{SPP}$, the more ordered and regular structures appear under laser irradiation. This value can be expressed as: (2)$$L_{SPP}=\frac{1}{2Im(\beta )}$$
where β is the SPP wave number given by [52]:(3)$$\beta =\frac{\omega }{c}\sqrt{\frac{\varepsilon_{air}\varepsilon_{m}}{\varepsilon_{air}+\varepsilon_{m}}}$$
where *c* is the speed of light, $\varepsilon_{air}$ is the dielectric permittivity of air, $\varepsilon_{film}$ is the dielectric permittivity of a medium.

For instance, for λ = 355 nm SPP free path in silver ($\varepsilon^{\prime }$= −2.0435 and $\varepsilon^{\prime \prime }$= 0.28156 [53]) $L_{SPPAg}∼$ 0.32 μm. Taking into account the critical value of free carrier density $N_{e}^{*}$ for TiO_2_ excitation, the dielectric constants of the film are $\varepsilon^{\prime *}$= −1 and $\varepsilon^{{*}^{\prime \prime }}$= 0.41951. Substituting these threshold values, we obtain $L_{SPP}^{*}$ = 0.034 μm, which is in order of magnitude smaller than for Ag film.

The experimentally obtained grating relief depth was estimated to be 20% of the film thickness, i.e., ∼35 ± 12 nm. In the regions of minima of the interference pattern, the surface level did not change significantly. Moreover, transmission electron microscopy revealed the increase of Ti and Ag concentration in the regions of interference maxima after laser exposure. Thus, the mechanism of the remnant relief recording is local densification of the porous film due to local laser heating.

Although it is difficult to determine the optimal depth of the relief due to the lack of information about the optical constants of the composite medium, for most cases, complete suppression of the reflection of laser radiation occurs when h∼λ/8 [54,55]. For 355 nm wavelength the estimated optimal depth is ∼44 nm, therefore the depth of the formed composite gratings is close to optimal.

## 5. Conclusions

In this article, we demonstrated the fabrication of highly ordered periodical nanocomposite gratings by direct UV laser writing. Formation of periodical gratings with the period of about d=λ/η = 327 ± 10 nm and DLOA δθ of 7–8${}^{\circ }$ occurred when laser fluence is relatively low, i.e., less than F = 31 mJ/cm^2^ and scan speed is about 50–250 mm/s (equal to 540–108 pulses per spot). Gratings formation is related to excitation of surface plasmon-polaritons on the boundary between the nanocomposite film and air under UV ns laser irradiation. The regularity of the gratings is supposed to be insured by the low free path of SPP in the composite material for the given wavelength. At a higher fluence range of up to 67 mJ/cm^2^ and a lower number of laser pulses per spot, two-dimensional periodical structures are formed. The grooves and ridges of the main relief were found to be modulated by the structures with a smaller period ∼D=λ/2ξ = 186 ± 8 nm. This grating is produced by the mutual interference of wedge and channel plasmon–polariton modes propagating in the opposite directions. However, this kind of 2D periodical gratings is less uniform and elongated. It is worth noting that the considered experimental phenomenon of laser-induced formation of polarization-dependent ordered periodic structures on TiO_2_:Ag film is a rare example of the universal polariton model realization under nanosecond laser exposure of dielectrics.

The obtained gratings exhibit optical anisotropy when viewed using crossed-polarizers. Not only the intensity of the transmitted and reflected light, but also spectral peaks depend on the orientation of the gratings relative to the polarizers axes. Such properties can be used for the production of planar optical elements, polarization elements and also for hidden security labels.

The proposed laser fabrication method provides faster, easier and more flexible patterning, compared to the conventional interference-based methods [17,18,19], as it allows the single-step self-organization of highly regular throughout nanocomposite gratings with a performance of about 0.4–0.9 cm^2^/min. The results show that by using commercially available single-beam scanning system instead of multiple-beam optical scheme, we can reduce the equipment’s complexity, ease the spatial tolerance requirements and produce large-area gratings simply by overlapping the irradiated areas.

## Figures and Tables

**Figure 1 nanomaterials-12-02811-f001:**
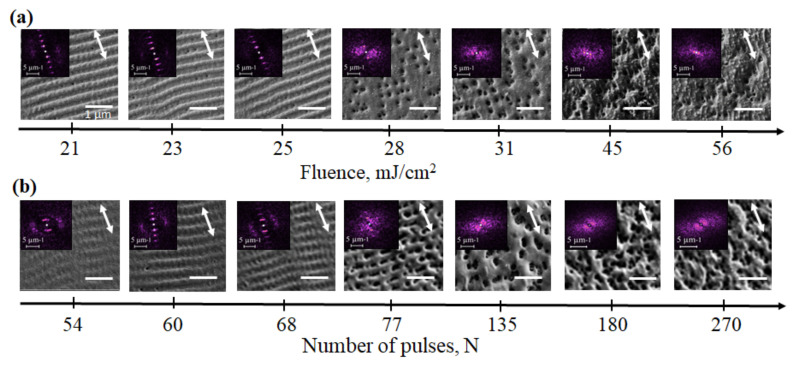
Evolution of periodical nanostructurs formation on TiO_2_:Ag film: (**a**) With the increase of power density for N = 540; (**b**) With the increase of N for F = 27 mJ/cm^2^. Double arrow is a direction of $\vec{E}$. Gratings with $\vec{g}\left|\right|\vec{E}$ and $\vec{G}⊥\vec{E}$ are identified for F = 21–25 mJ/cm^2^ and N = 54–68.

**Figure 2 nanomaterials-12-02811-f002:**
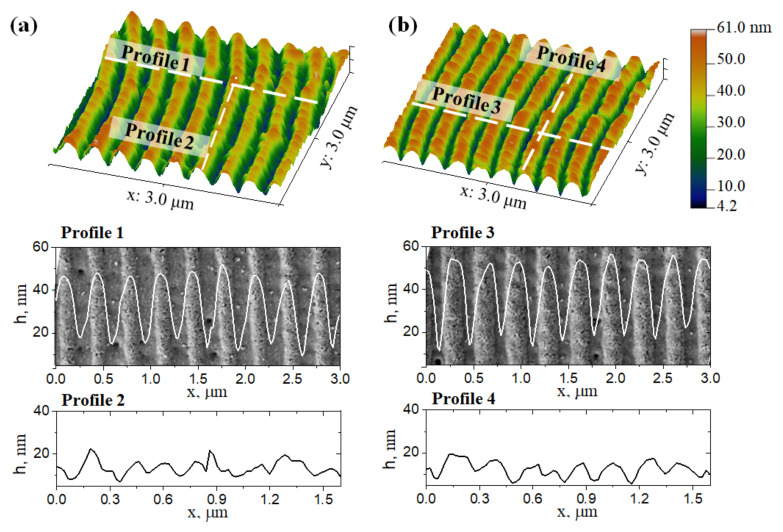
Surface topology of the periodic gratings obtained in various laser processing modes. Three-dimensional relief maps obtained using AFM and the corresponding surface relief profiles: (**a**) N = 540, F = 21 mJ/cm^2^; (**b**) N = 540, F = 23 mJ/cm^2^.

**Figure 3 nanomaterials-12-02811-f003:**
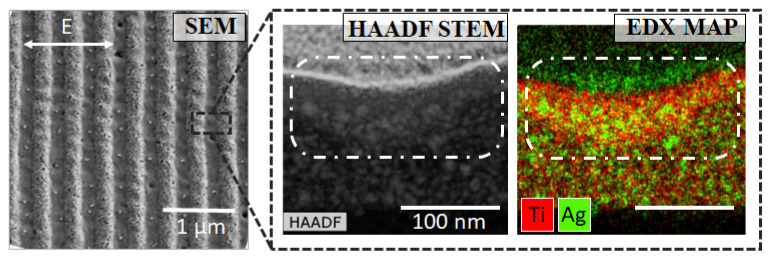
SEM, HAADF STEM and EDX map images of the structure (i) (N = 540, F = 21 mJ/cm^2^). Dash-dotted outline shows the densification area.

**Figure 4 nanomaterials-12-02811-f004:**
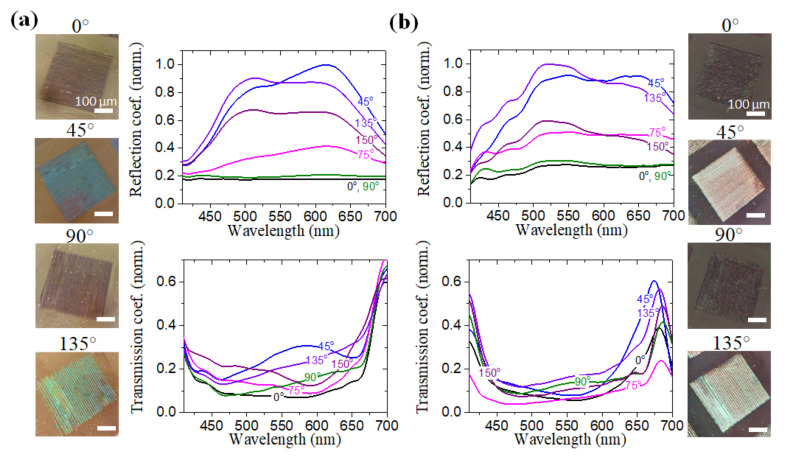
Optical properties of periodic nanocomposite gratings: microgimages of squares and reflection and transmission spectra for different rotation angles of periodic gratings in the crossed-polarizers: (**a**) structure (i) N = 540, F = 21 mJ/cm^2^; (**b**) structure (ii) N = 540, F = 23 mJ/cm^2^.

**Figure 5 nanomaterials-12-02811-f005:**
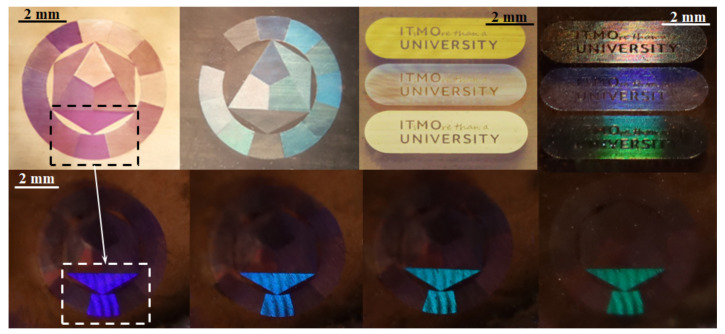
Examples of nanocomposite gratings applications: a color spectrum, where each sector was obtained with different laser processing parameters (transmission and reflection photos for different angles of illumination); ITMO University Logo—photos in reflection and scattering modes.

## Data Availability

Not applicable.

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
