# Peer review of "Laser Fabrication of Highly Ordered Nanocomposite Subwavelength Gratings"

_nanomaterials, 2022, doi:10.3390/nano12162811_

Round 1
Reviewer 1 Report
The paper “Laser fabrication of highly ordered nanocomposite subwavelength gratings” of Yaroslava Andreeva, et al., deals with optical nanogratings recorded at subwavelength scale. The focus is on the single-step fabrication process at UV wavelengths and the direct laser engraving in one or two dimensions. The results are well presented and the potential applications interesting. However, a very few things should be improved concerning the bibliography and the comparison with optical holography that allows similar results. Moreover, some minor adjustments are required as listed in the following:
1) Page 3 line 88, the Y axis is not defined or reported even in Fig. 1. Capital letter is not necessary here
2) Page 3 line 103, please define the grating vector
3) The grating vector should be reported on the images concerning sample morphology where it is possible
4) Can the Authors give an explanation about the shrinkage (ablation?) phenomenon observed in the grating area?
5) Page 6, Discussion section: It is very interesting the formation of structures perpendicular to the polarization direction. In the field of CW irradiation with only one beam other structures were observed always perpendicular to the used polarization (Riccardo Castagna, Daniele E. Lucchetta, Francesco Vita, Luigino Criante and Francesco Simoni, At a glance determination of laser light polarization state, Appl. Phys. Lett. 92, 041115 (2008))
6) Even if the scales are different: do authors think there could be a relationship between those two phenomena?
7) Concerning bibliography some more reference concerning holographic materials should be provided such as:
T. J. Bunning L. V. Natarajan V. P. Tondiglia and and R. L. Sutherland, Holographic Polymer-Dispersed Liquid Crystals (H-PDLCs), Annual Review of Materials Science 2000 30:1, 83-115, https://doi.org/10.1146/annurev.matsci.30.1.83
F. Vita, D. E. Lucchetta, R. Castagna, L. Criante and F. Simoni, 2009 J. Opt. A: Pure Appl. Opt. 11 024021
Yasuo Tomita, Naoaki Suzuki, and Katsumi Chikama, "Holographic manipulation of nanoparticle distribution morphology in nanoparticle-dispersed photopolymers," Opt. Lett. 30, 839-841 (2005)
- 8) Concerning the methods, it is important to underline/cite even other works in the field:
Haibin Zhang, Shane M Eaton, Jianzhao Li, Peter R Herman, “Femtosecond laser direct writing of multiwavelength Bragg grating waveguides in glass”, Opt. Lett., 2006 Dec 1;31(23):3495-7;
Y. Shimotsuma, P. G. Kazansky, J. Qiu, and K. Hirao, “Self-organized nanogratings in glass irradiated by ultrashort light pulses,” Phys. Rev. Lett. 24, 2474051-2474054 (2003);
E. Bricchi, J. D. Mills, P. G. Kazansky, B. G. Klappauf, and J. J. Baumberg, “Birefringent Fresnel zone plates in silica fabricated by femtosecond laser machining,” Opt. Lett. 27, 2200-2202 (2002).
9) Concerning the used approach, it appears to be time consuming and using expensive apparatus. Can Authors underline why this technique should be used when such subwavelength resolution is easily achievable by using two beams optical holography?

Author Response
We are thankful to the Reviewer for valuable comments. Please see the attachment with our point-by-point response.

Reviewer 2 Report
The manuscript nanomaterials-1861688 has been manly devoted to study a processing route for implementing particular nanoscale ordered nanocomposite gratings assisted by UV laser nanosecond pulses. Please see below a list of comments to the authors:
- A graphical abstract depicting the originality in the process of formation of the grating and what this paper adds to literature would be welcome.
- In order to highlight the value of the main results, the authors are invited to confront their results with gratings that can be obtained by versatile techniques assisted by multiple beams of irradiation. You can see for instance: http://dx.doi.org/10.1016/j.optlastec.2015.06.027
- Please justify the selection of the TiO2 in the study. SiO2 has been previously studied in a similar topic: https://doi.org/10.1016/j.optlaseng.2019.105840
- The authors employed Ag nanoparticles for the formation of the gratings. Is it expected a plasmonic or protoplasmonic response for the aim of the work? Please argue. A high-resolution of the nanoparticles would be important as an evidence of their shape and size.
- If possible, it would be useful to report the ablation threshold of the samples at this wavelength and pulse duration to discuss the modification in the fabrication process by a different laser source. You can see for instance: https://doi.org/10.3390/ma15134506
- In figure 2 is illustrated a non-monotonic inhomogeneous change in the peaks for the periodic grating. How is the influence of the Gaussian beam profile employed for these findings?
- Why the authors selected an UV wavelength for the irradiation if the localized surface plasmon resonance of the Ag is located in the visible region of the electromagnetic spectrum?
- In the conclusions the authors wrote about the gratings “Their formation is related to the surface plasmon-polariton excitation under UV laser irradiation” But the experiment corresponds to an off-resonance excitation. A better support for this asseveration ought to be included or verified.
- The preparation process of the nanocomposite films should be explained in the manuscript. Particularly, because reference 33 reports different nanoparticle distributions and the reproducibility of the submitted work is dependent on this condition.
- It is suggested to split the collective citations within the text. This is in order to easily visualize the importance of each selected citation to be part of the topic presented.
Author Response
We are grateful to the Reviewer for valuable comments and suggestions. Please see the attachment with our point-by-point response.

Round 2
Reviewer 2 Report
The authors have importantly improved the presentation of their main findings and most of the points have been clearly addressed in the reviewed version of the manuscript. In my opinion this work can be considered for publication in present form.